# Phosphoproteomic Analysis of the Jejunum Tissue Response to Colostrum and Milk Feeding in Dairy Calves during the Passive Immunity Period

**DOI:** 10.3390/ani13010145

**Published:** 2022-12-30

**Authors:** Xiaowei Zhao, Yunxia Qi, Tao Wu, Guanglong Cheng

**Affiliations:** Anhui Key Laboratory of Animal and Poultry Product Safety Engineering, Institute of Animal Science and Veterinary Medicine, Anhui Academy of Agricultural Sciences, Hefei 230031, China

**Keywords:** colostrum, calves, phosphoproteome, jejunum, Ti^4+^-immobilized metal ion affinity chromatography, liquid chromatography-tandem mass spectrometry

## Abstract

**Simple Summary:**

The jejunum tissue is an important site of nutrient digestion and absorption in neonatal calves. However, little is known regarding the effects of colostrum and mature milk intake on protein phosphorylation in the jejunal tissue of calves during the passive immunity period. Using titanium-immobilized metal ion affinity chromatography, coupled with the liquid chromatography-tandem mass spectrometry method, we found that the phosphoproteins were differentially expressed in the jejunum tissue of the neonate after ingested colostrum and milk feeding. The parental genes were primarily involved in developmental, glycohomeostasis, and immune regulation processes. We suggest that the changes induced in phosphorylated protein expression for calves fed with colostrum may play a key role in the developmental and innate immune process.

**Abstract:**

Improvements in the feeding of calves are of increasing importance for the development of the dairy industry. While colostrum is essential for the health of newborn calves, knowledge of protein phosphorylation alterations in neonatal calves that are fed colostrum or mature milk is lacking. Here, mid-jejunum tissue samples were collected from calves that received colostrum or milk. Subsequently, the jejunum phosphoproteome was analyzed using a phosphopeptide enrichment method, i.e., titanium immobilized metal ion affinity chromatography, coupled with liquid chromatography-tandem mass spectrometry. A total of 2093 phosphopeptides carrying unique 1851 phosphorylation sites corresponding to 1180 phosphoproteins were identified. Of the 1180 phosphoproteins, 314 phosphorylation sites on 241 proteins were differentially expressed between the groups. Gene ontology analysis indicated that the phosphoproteins were strongly associated with developmental and macromolecule metabolic processes, signal transduction, and responses to stimuli and insulin. Pathway analysis showed that the spliceosome, Hippo, insulin, and neurotrophin signaling pathways were enriched. These results reveal the expression pattern and changes in the function of phosphoproteins in bovine jejunum tissues under different feeding conditions and provide further insights into the crucial role of colostrum feeding during the early stages of life.

## 1. Introduction

Improving the feeding level of calves has become increasingly important for the healthy development of the dairy industry because it has a direct effect on the realization of their genetic potential in adulthood. Colostrum is essential for the health of newborn bovines because of its dual nutritional function and its basic nutritive substances, such as fat, protein, and minerals that can promote intestinal morphological development and functional maturity [1,2,3]. The major active nutraceutical ingredients, such as immunoglobulins, can help neonates establish passive immunity [4,5]. Therefore, the first meal of colostrum is generally recognized as the most important event for newborn calves [6,7], and colostrum quality and management can have a pronounced effect on overall dairy calf health [8,9].

The small intestine is the main organ involved in the digestion and absorption of colostral components [5,10]. It is an important immune organ due to its function as an immune barrier, which protects the host from invasion by pathogenic microorganisms and maintains intestinal health [11,12]. The growth-promoting effects of colostrum in the bovine small intestine have been studied extensively, and the development and application of omics techniques have provided new insights into the crucial role of colostrum feeding in calves during the early stages of life. A previous study used two-dimensional gel electrophoresis-based proteomic analysis to study the changes in intestinal proteins induced by colostrum uptake and found that IgG and β-lactoglobulin were internalized within the small intestine [13]. Metabolomic studies have shown that components of colostrum can enter the circulatory system to meet the needs of various tissues or cells, after being transported across the intestinal epithelium [14,15,16]. In addition, transcriptome analysis revealed changes in gene expression contributing to intestinal health and development after colostrum ingestion and absorption [17,18,19]. These studies suggest that the expression of several proteins, metabolites, or genes in the neonatal intestine may contribute to the metabolism and absorption of colostral components. In our recent study of liver tissue in newborn calves, differentially expressed phosphorylated proteins identified after colostrum feeding were found to be involved in energy metabolism and immune response processes [20]. In addition, a quantitative phosphoproteomics study of the bovine small intestine during the first 36 h of life showed that the phosphorylated proteins were regionally and temporally expressed across different developmental time points. These proteins were involved in vesicle-mediated transport, tight junction formation, and immune response [21], indicating that protein phosphorylation may play a key role in regulating several biological processes. However, knowledge of the effects of colostrum and mature milk intake on protein phosphorylation in the jejunal tissue of calves during the passive immunity period is still lacking.

We hypothesized that the phosphoproteomics of jejunal tissue would differ after newborn calves received colostrum in comparison to mature milk and that differentially expressed phosphoproteins could be related to multiple functions. In this study, we aimed to identify and quantify phosphoproteins in the jejunal tissue of newborn calves subjected to colostrum or mature milk feeding using Ti^4+^-immobilized metal ion affinity chromatography (Ti^4+^-IMAC). We anticipated that our results could facilitate the characterization of protein phosphorylation changes and their potential regulatory functions in jejunal tissue during the passive immunity period under different nutritional strategies.

## 2. Materials and Methods

### 2.1. Experimental Design and Sample Collection

This study was approved by the Ethical Committee of the Anhui Academy of Agricultural Sciences (A11-CS16). Six healthy male Holstein calves from multiparous dairy cows (2–4 parity) were weighed (40 ± 2 kg) and randomly assigned to two groups: (1) colostrum feeding group (CF, n = 3), and (2) mature milk feed group (MF, n = 3). Calves were recruited and separated from their dams immediately after birth and placed in individual Calf-Tel (Germantown, WI, USA) hutches. Colostrum (the first two milking colostrum) and mature milk were obtained from healthy multiparous Holstein cows and stored at −20 °C. All milk samples were heated to 40 °C in a water bath prior to use. Using nipple bottles, each calf was fed 2 L of colostrum or mature milk two times within 2 h and at 12 h after birth. Mid-jejunum samples were collected after calves were euthanized at 24 h of birth and all samples were placed in liquid nitrogen immediately, transported to the laboratory, and stored at −80 °C before protein extraction.

### 2.2. Protein Extraction and Digestion

Protein was extracted as previously described [20]. In brief, a pestle and mortar were used to grind the jejunum tissue samples into powder. The samples were then homogenized in ten volumes of tissue protein lysis buffer (8 M urea, 2 M thiourea, 4% 3-[(3-cholamidopropyl)-dimethylammonium]-1-propanesulfonate (CHAPS), 20 mM Tri-base, 30 mM dithiothreitol) using a high-intensity ultrasonic processor (Scientz, Ningbo, China). Three volumes of acetone were added to the solution, and the precipitate was collected after centrifuging the mixture at 13,000× *g* and 4 °C for 20 min. The Bradford method was used to determine the protein concentration of the precipitate dissolved in 100 μL of 5 M urea. Approximately 800 μg protein solution was incubated with 100 mM dithiothreitol for 1 h, followed by 100 mM iodoacetamide for 1 h in the dark. In the final step, samples were digested at 37 °C overnight with sequencing-grade modified trypsin [enzyme: protein (W:W = 1:50)]. The dried tryptic peptide samples were stored at −80 °C for phosphopeptide enrichment.

### 2.3. Phosphopeptide Enrichment Using Ti^4+^-IMAC

Ti^4+^-IMAC materials were prepared according to the method described in our previous study [20]. To enrich the phosphopeptides, the tryptic peptide mixtures were first incubated with a Ti^4+^-IMAC microsphere (5 mg) suspension with vibration in 500 μL of binding buffer (6.0% trifluoroacetic acid, TFA; 80% acetonitrile, ACN) at room temperature (RT) for 120 min. Then the mixture was centrifuged at 13,000× *g* and RT for 5 min, and the captured phosphopeptide beads were washed sequentially with 1 mL of 0.6% TFA, 50% ACN, and 200 mM NaCl, followed by 1 mL 0.1% TFA and 30% ACN. Subsequently, the phosphopeptides were eluted twice with 100 μL of 500 mM dipotassium phosphate (pH 7.0), and Zip-Tip C18 columns (Millipore, Burlington, MA, USA) were used for desalted pooled elutions. After collection, the supernatant with phosphopeptides was vacuum dried for subsequent liquid chromatography-tandem mass spectrometry analysis (LC-MS/MS).

### 2.4. Phosphopeptide Identification and Phosphorylation Site Localization Using LC-MS/MS

Phosphopeptides were redissolved in 100 μL 0.1% formic acid and centrifuged at 13,000× *g* and 4 °C for 15 min to collect their supernatants. The peptides were injected into an Easy C18 column (100 μm I.D. × 20 mm, 5 μm) using an autosampler, and each sample was analyzed in triplicate. The peptides were analyzed using an EASY-nLC 1000 coupled Q-Exactive mass spectrometer (Thermo Fisher Scientific, Waltham, MA, USA). The samples were loaded onto a trap column (75 μm inner diameter; fused silica filled with 3.0 μm Aqua C18 beads; Thermo Fisher Scientific) in buffer A (0.1% formic acid) and buffer B (0.1% formic acid in acetonitrile) at a flow rate of 0.35 μL/min. A gradient of 3–90% buffer B was applied for 120 min, followed by a 10-min gradient of 90%. Data-dependent ion signals were collected and run with the following settings: resolution 70,000; automatic gain control target, 3 × 10^6^; maximum injection time, 20 ms; scan range, *m*/*z* 300–1800. The resolution of the MS/MS was 17,500, and the loop count was 10. Charge exclusion: 1, >8, and dynamic exclusion: 20 s. Thermo Fisher Scientific Xcalibur software version 2.2 was used for processing raw data.

### 2.5. Phosphoprotein Identification, Site Localization, Peptides Analysis, and Quantification

Raw MS/MS data were retrieved from a database containing 46,498 protein entries from *Bos taurus* (UniProt database) using MaxQuant software (version 1.6.1.0, Max Planck Institute of Biochemistry, Munich, Germany). The search parameters were as follows: enzyme, trypsin/P; max missed cleavage, 2; fixed modification, carbamidomethylation (C); variable modification, oxidation (M), and phosphorylation (Ser/Tyr/Thr). An analysis of decoy databases was conducted with a false discovery rate of 0.01 to filter false positive identification at the protein and peptide levels. Mass tolerances for precursors and fragments were set at 20 ppm and 4.5 ppm, respectively. After phosphoprotein identification and quantitation, data filtering was conducted using Perseus software. Phosphorylation sites with a probability cutoff of >0.75 were considered localized. At least two valid values were required in at least one group for each of the identified peptides, with three replicates in each group. Label-free quantification was performed, and changes in the expression level of phosphoproteins were considered significant at *p* ≤ 0.05. A model of modified 13-mers in all protein sequences was analyzed using the WebLogo software (http://weblogo.berkeley.edu, accessed on 17 March 2022). Based on the characteristics of the surrounding residues, kinase motifs were divided into three classes: pro-directed, acidic, and basic, based on the detailed method described in Villén et al. [22].

### 2.6. GO and KEGG Pathway Analysis

The DAVID online tool (https://david.ncifcrf.gov, accessed on 25 July 2022) was used to enrich pathways and terms related to Gene Ontology (GO). The significantly enriched GO terms and KEGG pathways were ranked by enrichment scores. Protein–protein interaction networks were constructed with medium confidence (0.40 P) using STRING online tool (https://string-db.org, accessed on 29 July 2022) and Cytoscape software (version 3.6.0, National Resource for Network Biology).

## 3. Results

### 3.1. Phosphoproteome Profiles of Jejunum Tissue Response to Colostrum and Mature Milk Feeding

To investigate phosphoprotein abundance, jejunum tissues were harvested from six neonatal male calves that had ingested colostrum (CF; n = 3) or mature milk (MF; n = 3). Proteins were digested in solution after tissue homogenization, followed by strong cation exchange chromatography and Ti^4+^-IMAC phosphopeptide enrichment. A total of 1180 phosphoproteins were detected in the jejunum tissue with 1011 and 1028 phosphoproteins identified in the CF and MF groups, respectively (Figure 1a and Appendix A). Among the identified phosphoproteins, the two treatments shared 859 phosphoproteins; 152 phosphoproteins were unique to the CF group, and 169 phosphoproteins were only identified in the MF group (Figure 1b).

### 3.2. Characterization of the Phosphopeptides and Phosphorylation Sites of Phosphoproteins

From the 1180 phosphoproteins detected in jejunum tissue, a total of 2093 phosphopeptides carrying 1851 unique phosphorylation sites were identified. Of those, 1734 and 1742 phosphopeptides containing 1505 and 1557 phosphorylation sites were detected in the CF and MF groups, respectively. The relative abundance of phosphorylation sites of serine (Ser), threonine (Thr), and tyrosine (Tyr) residues were analyzed and accounted for 90.3%, 9.3%, and 0.4%, respectively (Figure 1c). In addition, phosphosites that belong to the pro-directed, basic, acidic, and tyr-directed motif categories accounted for 58.5%, 18.0%, 15.9%, and 1.1%, respectively (Figure 1d). More than ten kinase motifs were identified, including two pro-directed motifs (S-P and T-P), four acidophilic kinase motifs (S-D, S-E, S-x-E, and S-x-D), two basic motifs (R-x-x-S and K-x-x-S), and two motifs (S-x-P and S-x-S) that were not related to particular kinases (Figure 2).

### 3.3. Quantitation of Protein Phosphorylation Changes

We first evaluated the biological function of phosphoproteins identified in the jejunal tissue of newborn calves, and 1011 phosphoproteins were strongly enriched in tight junctions, focal adhesion, endocytosis, insulin signaling pathway, regulation of actin cytoskeleton, and the mTOR signaling pathway (Appendix A). To further analyze the functions of the specific phosphoproteins, differentially expressed phosphoproteins were compared using a label-free quantitative phosphoproteomic method. A total of 314 phosphorylated sites derived from 241 proteins were significantly altered between the two study groups based on *p* < 0.05 and log2 fold change ≧1.5 (Appendix A). To visualize the distribution of differentially expressed phosphoproteins, volcano plots were plotted between the CF and MF groups (Figure 3) using the Origin software (version b9.5.0.193, OriginLab, Northampton, MA, USA). Of the 241 differentially expressed phosphoproteins, 206 phosphorylation sites on 153 proteins were upregulated and 108 phosphorylation sites on 101 proteins were downregulated in the CF group. The upregulated phosphoproteins in the CF compared with the MF group included glycogen synthase kinase-3 beta (GSK3β), thyroid hormone receptor interactor 10 (TRIP10), epidermal growth factor receptor pathway substrate 8 (EPS8), phosphoglucomutase-1 protein (PGM1), and eukaryotic translation initiation factor 4 B (EIF4B). The downregulated phosphoproteins in the CF compared with the MF group were TSC complex subunit 2 (TSC2), interleukin 1 receptor-associated kinase 3 (IRAK3), 2-hydroxyacyl-CoA lyase 1 (HACL1), and tyrosine-protein phosphatase non-receptor type (PTPN1).

### 3.4. Functional Categorization of Phosphoproteins from Gene Ontology (GO) Terms

To investigate the potential function of protein phosphorylation in the jejunal tissue of calves under different feeding strategies, GO term and Kyoto Encyclopedia of Genes and Genomes (KEGG) pathway enrichment analyses were conducted. Functional classification showed that most differentially expressed phosphoproteins were mainly enriched in macromolecule metabolic process, response to stimulus, developmental process, and signal transduction. The primary cellular component terms were membrane-bounded organelle, cytoplasm, nucleus, macromolecular complex, and cytoskeleton. For molecular function, the most enriched terms were protein binding, enzyme binding, macromolecular complex binding, kinase activity, and structural molecule activity. KEGG pathway analysis indicated that these differentially expressed phosphoproteins were associated with the spliceosome, Hippo, insulin, and neurotrophin signaling pathways (Figure 4 and Appendix A). Due to the critical role of insulin in the regulation of growth in neonates, particular attention has been given to the insulin signaling pathway. Many phosphoproteins involved in the insulin signaling pathway including the B-Raf proto-oncogene, serine/threonine kinase (BRAF), TSC2, GSK3β, TRIP10, and PTPN1 were identified (Figure 5).

### 3.5. Phosphoprotein–Phosphoprotein Interaction Networks

To further examine the interconnection of phosphoproteins in jejunal tissue during the passive immunity period, protein–protein interaction analysis of differentially expressed phosphoproteins between the CF and MF groups was performed. We found that 99 proteins with 280 interactions were linked in the network and 20 differentially expressed phosphoproteins interacted with more than 10 proteins, including the U2 snRNP auxiliary factor large subunit (U2AF2), heterogeneous nuclear ribonucleoprotein D0 (HNRNPD), serine/arginine repetitive matrix protein 1 (SRRM1), EIF4B, and GSK3β. U2AF2 and HNRNPD had the most connections due to their interactions with 18 proteins in the network (Figure 6).

## 4. Discussion

Calf health and survival are highly dependent on colostrum feeding, especially during the passive immunity period. Colostrum components are digested and absorbed across the digestive tract to meet the needs of growth and development [24]. In this study, we aimed to explore the expression pattern of the phosphoproteome in the jejunum of newborn calves in response to colostrum and mature milk feeding during the passive immunity period. In total, 1851 phosphorylation sites on 2093 peptides corresponding to 1180 proteins were identified in jejunal tissue. Phosphosites belonging to the pro-directed motif category were the most abundant (58.5%), followed by the basic (18.0%), acidic (15.9%), and tyr-directed motifs (1.1%). Based on phosphoproteomic quantitative analysis, 241 differentially expressed phosphorylated proteins with 314 phosphorylated sites were detected in the CF and MF groups. We revealed that the differentially expressed phosphoproteins modulated by colostrum and mature milk intake were related to the macromolecule metabolic process, response to stimulus, developmental process, signal transduction, and response to insulin biological process. Early phosphorylation events are associated with the spliceosome, Hippo, insulin signaling, and neurotrophin signaling pathways. Our results showed the phosphoproteomic profile and its potential regulatory role in jejunum tissue under different nutritional strategies, providing novel insights into the importance of colostrum feeding in newborn calves during the first 24 h of life.

### 4.1. Phosphoproteins Changes Are Associated with Growth and Development

In the present study, differentially expressed phosphoproteins were shown to be involved in various biological functions according to bioinformatics analysis of their predicted target genes. GO analysis indicated that the phosphoproteins influenced by colostrum and mature milk feeding were associated with the developmental process. Previous studies have shown that colostrum components can be transported across the intestine in neonatal calves [14,25], serving as a unique source of nutrients and bioactive components that promote intestinal development. We found that the phosphorylation levels of several proteins were upregulated in calves that received colostrum treatment. For example, mammalian tissues universally express two isoforms of GSK3, which is evolutionarily conserved in intracellular serine-threonine kinase [26]. The phosphorylation level of GSK3β (Ser216) was found to be increased in calves that received colostrum. A previous study showed that serine phosphorylation of GSK3β was positively related to embryo development [27], mainly through its participation in the Wnt signal transduction pathway [28]. Notably, growth retardation effects were observed in GSK3β KO juvenile mice, which exhibited inefficient breathing patterns at the late stages and failed to survive [29]. An additional phosphoprotein EIF4B (Ser422) has been shown to be upregulated in colostrum-fed calves and plays an important role in cell survival and proliferation [30]. EIF4B is phosphorylated at Ser422 by S6K in response to insulin in a rapamycin-dependent manner [31], which is consistent with our results. In insects, EIF4B overexpression stimulates the proliferation of Drosophila cultured cells and eye imaginal discs, in contrast, *EIF4B* siRNA inhibited basic translation and affected cell survival [32]. The RNAi-mediated silencing of *EIF4B* results in polysome depletion and translational repression in mammalian cells [33]. Interestingly, we found that the phosphorylation of TSC was downregulated when calves ingested colostrum. Several mutations in TSC2 and TSC1 cause tuberous sclerosis complex [34]. Normally, TSC2 and TSC1 can form a physical and functional complex and inhibit the phosphorylation of RHEB. RHEB is necessary for stimulating the phosphorylation of mTOR and plays an essential role in the regulation of S6K and 4EBP1 in response to nutrients and cellular energy status, thus regulating cell growth [35]. Overall, this suggests that the altered phosphoproteins in jejunal tissue may contribute to the developmental regulation of newborn calves during the early stages of life.

### 4.2. Phosphoproteins Associated with Insulin Signaling Pathway Play an Important Role in Glycometabolism

In newborn ruminants, the glucose supply from the placenta translates into lactose supply [36]. As lactose uptake does not meet the glucose demand in neonates, gluconeogenesis is considered an important metabolic pathway for establishing postnatal euglycemia [37]. Furthermore, blood insulin concentration has been found to increase in neonatal calves that received colostrum [24]. As expected, we found that several phosphoproteins involved in the insulin signaling pathway were upregulated in the calves that had ingested colostrum in the current study, including TRIP10, also known as Cdc42-interacting protein-4 (CIP4), which is an adaptor protein involved in numerous cellular processes depending on the lineage and tissue [38]. TRIP10 regulates insulin-induced glucose transporter 4 (GLUT4) translocation to the plasma membrane in adipocytes via interaction with TC-10 [39,40]. Interestingly, TRIP10 increases endocytosis of GLUT4 in muscle cells via bidirectional interaction with N-WASp and dynamin-2, thereby inhibiting glucose uptake [41,42]. PGM1, an active member of the PGM enzyme family, plays a role in several metabolic pathways, such as glycolysis, glycogenesis, and N-linked glycosylation [43]. Several studies have shown that glucose 1-phosphate and glucose 6-phosphate are interconverted by PGM1 [44,45,46], and the absence of PGM1 causes a reduction in plasma glucose levels [47], suggesting that PGM1 is responsible for maintaining cellular glucose homeostasis. Therefore, elevated levels of these phosphoproteins appear to contribute to improving the regulation of glycogen transport and homeostasis in neonatal calves.

### 4.3. Phosphorylated Proteins Induced by Colostrum Are Involved in the Innate Immunity Response

In addition to the digestion and absorption of nutrient components, the small intestine is an important immune organ and provides innate immunity that acts as the first line of defense to resist the invasion of pathogenic microorganisms before passive immunity becomes functional and established. EPS8, which is involved in the immune process, was upregulated in the colostrum-fed calves in our study. An EPS8 protein contains a pleckstrin homology domain, a Src homology 3 domain, and several proline-rich regions [48,49]. A previous study showed that EPS8 could be induced by lipopolysaccharide (LPS)-treated macrophages, and this induction increased the phagocytic activity and bacterial killing effect of macrophages. However, LPS-induced TLR4/MyD88 complex formation was damaged in *EPS8* KO mice, impairing the *E. coli*-killing ability of the macrophages [50]. We found that the phosphorylation of IRAK3 was downregulated when calves ingested colostrum. Interleukin-1 receptor-associated kinases (IRAKs) play a crucial role in innate immune signaling that mediates the host’s response to pathogens. IRAK3 is an isoform of the IRAKs that belongs to the serine-threonine kinase family and is widely expressed in different tissues and immune cells. A previous study suggested that IRAK3 is a negative signal regulator, and a higher inflammatory response to bacterial infection and increased susceptibility to LPS-induced septic shock are observed in *IRAK3*−/− mice [51]. There is evidence that IRAK3 is a negative regulator of innate immunity based on clinical data, as multiple mutations in IRAK3 were linked to the early onset persistent asthma pathogenesis [52]. Additionally, an increase in IRAK3 and mutations in the pseudokinase domain are correlated with an increased risk of many diseases [53,54]. In summary, these colostrum intake-associated alterations in phosphoprotein profiles in jejunal tissue may be beneficial for the development and overall health of calves during the early stages of life.

### 4.4. GSK3β-TSC2 Interaction Regulates Biological Process through the mTOR Signaling Pathway

A protein–protein interaction network was constructed to further analyze the differentially expressed phosphoproteins in the jejunal tissues of calves that ingested colostrum or mature milk. In the protein–protein interaction network, GSK3β interacted with TSC2 (Figure 6), and we observed that the phosphorylation level of GSK3β was upregulated and the phosphorylation level of TSC2 was downregulated in the jejunal tissue after calves received colostrum (Figure 3). Previous studies have shown that TSC2 is an upstream negative regulator of mTOR and has potential GSK3β phosphorylation sites [55]. By phosphorylating AMPK and GSK3β, TSC2 could regulate cell growth by integrating Wnt and energy signals [56]. Therefore, our results confirmed that one protein can affect a range of biological processes by interacting with other proteins; however, such changes in jejunal protein–protein interactions require further research.

## 5. Conclusions

In this study, we aimed to explore the expression pattern of the phosphoproteome in the jejunum of newborn calves in response to colostrum and mature milk feeding during the passive immunity period. The results showed the phosphoproteomic profile and its potential regulatory role in jejunum tissue under different nutritional strategies, which will provide novel insights into the importance of colostrum feeding in newborn calves during the passive immunity period.

## Figures and Tables

**Figure 1 animals-13-00145-f001:**
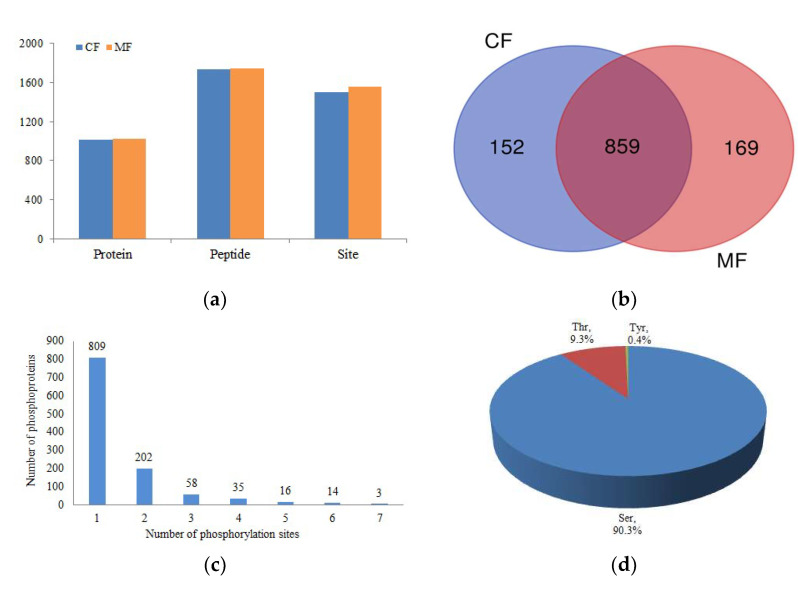
Phosphoproteomic analysis overview of the jejunum tissues of newborn calves fed colostrum and mature milk. (**a**) Number of phosphoproteins, phosphopeptides, and phosphosites identified in CF and MF. (**b**) Number of unique phosphoproteins identified in CF and MF. (**c**) Distribution of phosphorylation sites in the phosphoproteins. (**d**) Distribution of the phosphorylated amino acids.

**Figure 2 animals-13-00145-f002:**
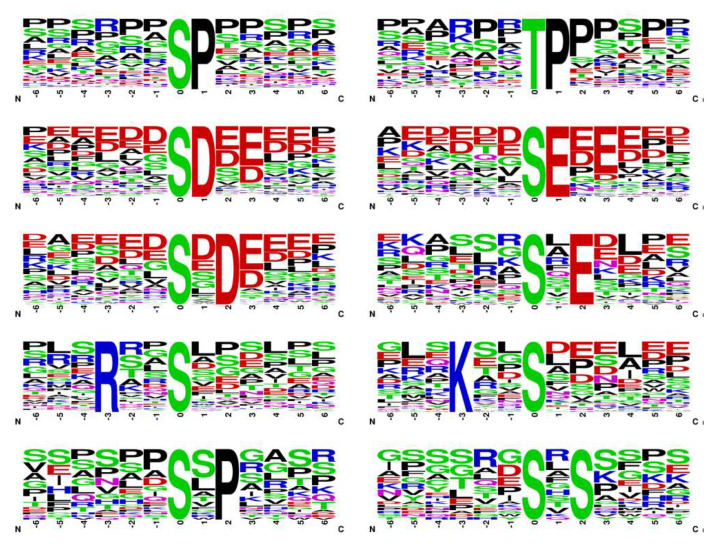
Kinase motifs extracted from the jejunum tissue of newborn calves. The height of each letter represents the frequency of the specific AA residue at that position.

**Figure 3 animals-13-00145-f003:**
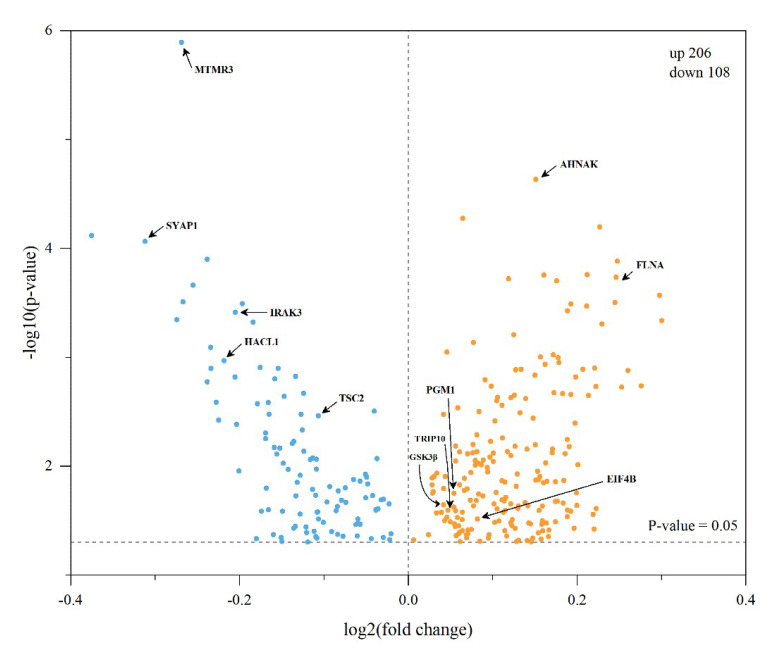
Regionally detected differentially expressed phosphoproteins in jejunum tissue. The X and Y axes show log^2^ (fold change) and –log^10^
*p*-value of each differentially expressed phosphoprotein, respectively. Saffron yellow and blue spots indicate upregulated and downregulated, respectively.

**Figure 4 animals-13-00145-f004:**
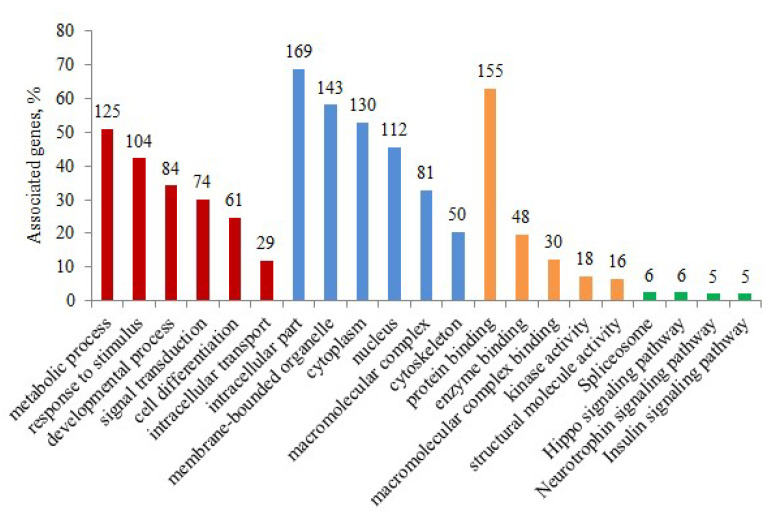
GO and KEGG pathway analysis for differentially expressed phosphoproteins. The number of genes is represented by bar height. The ordinate represents the percentage of identified genes compared to all the genes. The abscissa corresponds to the GO terms or KEGG pathways. See also Appendix A. Red, blue, orange, and green indicates biological process, cellular component, molecular function, and KEGG pathway, respectively.

**Figure 5 animals-13-00145-f005:**
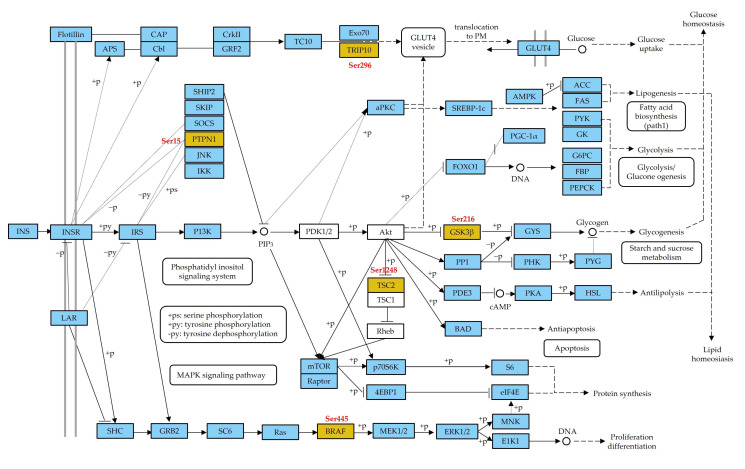
Phosphoproteins of the insulin signaling pathway identified in the jejunum tissues of newborn calves. Orange box indicates the proteins identified in this study. Red letter above or near the protein indicates the phosphosites of this protein identified in this study. KEGG pathway [23] was redrawn with the Kanehisa laboratories permission.

**Figure 6 animals-13-00145-f006:**
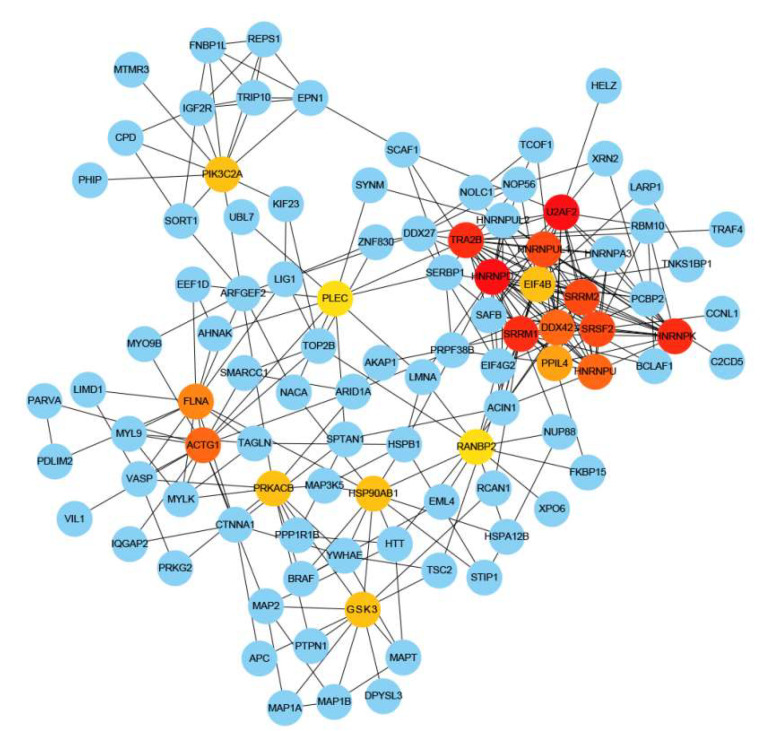
A protein−protein interaction network constructed using the STRING online database and Cytoscape software. Nodes represent phosphoproteins, edges represent interactions between proteins. Nodes in red, orange or yellow indicate a higher number of interactions, blue nodes indicate phosphoproteins with fewer interactions.

## Data Availability

Not applicable.

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
