# Peer review of "Phosphoproteomic Analysis of the Jejunum Tissue Response to Colostrum and Milk Feeding in Dairy Calves during the Passive Immunity Period"

_animals, 2022, doi:10.3390/ani13010145_

Round 1

Reviewer 1 Report

Comments on the methodology of the manuscript:

1.    Why were only 3 calves per group planned in the comparative studies instead of 4-5 animals?

2.    Please describe to what degree the biological material was aligned (e.g. Did the calves have the same father?).

3.    Please describe:

-          the diet of the mothers,

-          age of cows

-          calving term (month),

-          Did the calves receive colostrum of their mothers or pooled colostrum of 6 cows?

-          Did the mature milk come from one cow or several?

Author Response

We must thank you and reviewers critical comments and thoughtful suggestions. We have made careful modifications on the original manuscript. All changes were marked in yellow text. Below you would find our point-by-point responses to your comments/questions:

Reviewer: 1
Comments on the methodology of the manuscript:

  1. Why were only 3 calves per group planned in the comparative studies instead of 4-5 animals?

AU: How many replicates are required to obtain reliable results? This question is normally answered by a Power analysis. A Power analysis requires a desired fold change and an estimation of the variability of the data. Intuitively, every protein has a unique variance, and this variance might be different for different conditions. For this reason, in a proteomics dataset a Power analysis can be performed, but the ideal number of replicates is different for every protein (Levin, 2011). A power analysis can be used to calculate the minimum number of biological replicates required for a study. The power was set to 0.8 and confidence level to 0.05, a variation of 20% would require a minimum fold change of 2, which includes three biological replicates per group. In many previous studies, three biological replicates were applied in per group (Liang et al., 2014; Liang et al., 2016). In our study, three biological replicates per group were referred the previous studies, and Power analysis was not performed.

References:

Levin, Y. The role of statistical power analysis in quantitative proteomics. Proteomics, 2011, 11(12):2565-2567.

Liang G, Malmuthuge N, McFadden TB, Bao H, Griebel PJ, Stothard P, Guan le L. Potential regulatory role of microRNAs in the development of bovine gastrointestinal tract during early life. PLoS One. 2014 Mar 28;9(3):e92592.

Liang G, Malmuthuge N, Bao H, Stothard P, Griebel PJ, Guan le L. Transcriptome analysis reveals regional and temporal differences in mucosal immune system development in the small intestine of neonatal calves. BMC Genomics. 2016 Aug 11; 17(1): 602.

  1. Please describe to what degree the biological material was aligned (e.g. Did the calves have the same father?).

AU: Thank you for your reminding, cows are mated by artificial insemination, and all calves used in this study have the same father.

  1. Please describe:

- the diet of the mothers,

AU: All mothers are ingested the same diet (consist of corn, soybean, Oat Hay, whole corn silage, etc.) in the dry period, and used the same feeding management. So, we do not added this information in the manuscript, if necessary, we will add relate content.

  • age of cows

AU: The age of cows were 2-4 parity, the necessary information was added.

- calving term (month),

AU: All calves were birth at November.

Did the calves receive colostrum of their mothers or pooled colostrum of 6 cows? -   Did the mature milk come from one cow or several?

AU: The first 2 milking colostrum samples were taken from healthy, multiparous Holstein dairy cows and pooled, and bulk milk from a healthy herd of experimental dairy farm cows was collected, placed in plastic bottles. This have been described in our previous study (Zhao et al., 2018)

References:

Zhao, X. W., Qi, Y. X., Huang, D. W., Pan, X. C., Cheng, G. L., Zhao, H. L., & Yang, Y. X. (2018). Changes in serum metabolites in response to ingested colostrum and milk in neonatal calves, measured by nuclear magnetic resonance-based metabolomics analysis. Journal of dairy science, 101(8), 7168–7181. https://doi.org/10.3168/jds.2017-14287

Reviewer 2 Report

The manuscript "Phosphoproteomic analysis..." is well-written and relevant to the field.

I have just a few observations:

1) The authors should clarify the choice of time of 24 for the analyses.

2) Authors should give more details about jejunum samples. Which segment of the jejunum was obtained? Was the part close to the ileum? What is the size or weight of this segment? How long did it take between slaughter and the samples being placed at -80ºC the samples were obtained?

3) Figure 6 should be improved. It's hard to read the letters inside the circles.

Author Response

We must thank you and reviewers critical comments and thoughtful suggestions. We have made careful modifications on the original manuscript. All changes were marked in yellow text. Below you would find our point-by-point responses to your comments/questions:

Comments on the methodology of the manuscript:

The manuscript "Phosphoproteomic analysis..." is well-written and relevant to the field.

I have just a few observations:

1) The authors should clarify the choice of time of 24 for the analyses.

AU: The first 24 h is essential for newborn calve to establish passive immunity, previous studies showed that the ability of small intestine to take up IgG is the strongest within 6 hours after birth, decreases from 6-12 hours, and nearly stops at 24 hours after birth (Moore et al., 2005; Cabral et al., 2012). Thus, we choice the time of 24 for this analyses.

References:

Moore M, Tyler JW, Chigerwe M, Dawes ME, Middleton JR. Effect of delayed colostrum collection on colostral IgG concentration in dairy cows. J Am Vet Med Assoc, 2005, 226(8): 1375-1377.

Cabral RG, Kent EJ, Haines DM, Erickson PS. Addition of sodium bicarbonate to either 1 or 2 feedings of colostrum replacer: effect on uptake and rate of absorption of immunoglobulin G in neonatal calves. J Dairy Sci, 2012, 95(6): 3337-3341.

2) Authors should give more details about jejunum samples. Which segment of the jejunum was obtained? Was the part close to the ileum? What is the size or weight of this segment? How long did it take between slaughter and the samples being placed at -80ºC the samples were obtained?

AU: Thanks for your reminding. In this study, mid-jejunum samples were collected after calves were euthanized at 24 h of birth and all samples were first snap-frozen in liquid nitrogen, and transported to the laboratory, stored at −80 °C until analysis.

All necessary were revised in the manuscript.

3) Figure 6 should be improved. It's hard to read the letters inside the circles.

AU: Thanks for your reminding and the representation of Figure 6 was revised in the manuscript.

Reviewer 3 Report

Authors conducted a trial with 6 calves, 3 of them fed with colostrum and the others with mature milk during the first day of their life. Phosphoproteome of jejunum was analyzed by affinity chromatography, coupled with liquid chromatography-tandem mass spectrometry to reveal differentially expressed markers.

Comments to authors:

L85: Correct the wieght of newborn calves.

Fig 3: Describe the meaning of yellow and blue spots.

Fig 6: What do you mean darker colour in the figure legend?

Author Response

We must thank you and reviewers critical comments and thoughtful suggestions. We have made careful modifications on the original manuscript. All changes were marked in yellow text. Below you would find our point-by-point responses to your comments/questions:

Comments to authors:

L85: Correct the wieght of newborn calves.

AU: Thanks for your reminding, and weighed (40 ± 2 g) was revised as weighed (40 ± 2 kg).

Fig 3: Describe the meaning of yellow and blue spots.

AU: Thanks for your reminding, in fig 3, the saffron yellow indicate upregulated, the blue spots indicate downregualted. All necessary were revised in fig 3.

Fig 6: What do you mean darker colour in the figure legend?

AU: Revised, the previous expression is inaccuracy. Nodes in red, orange, or yellow indicate a higher number of interactions, blue nodes indicate phosphoproteins with fewer interactions. All necessary were revised in fig 6.

Round 2

Reviewer 1 Report

I accept the answers to my questions/comments.